# Neural structure mapping in human probabilistic reward learning

**Fabrice Luyckx[1]\*, Hamed Nili[1,2], Bernhard Spitzer[1,3†], Christopher Summerfield[1†]\***

[1]Department of Experimental Psychology, University of Oxford, Oxford, United Kingdom; [2]Wellcome Centre for Integrative Neuroimaging, University of Oxford, Oxford, United Kingdom; [3]Center for Adaptive Rationality, Max Planck Institute for Human Development, Berlin, Germany

**Abstract** Humans can learn abstract concepts that describe invariances over relational patterns in data. One such concept, known as magnitude, allows stimuli to be compactly represented on a single dimension (i.e. on a mental line). Here, we measured representations of magnitude in humans by recording neural signals whilst they viewed symbolic numbers. During a subsequent reward-guided learning task, the neural patterns elicited by novel complex visual images reflected their payout probability in a way that suggested they were encoded onto the same mental number line, with 'bad' bandits sharing neural representation with 'small' numbers and 'good' bandits with 'large' numbers. Using neural network simulations, we provide a mechanistic model that explains our findings and shows how structural alignment can promote transfer learning. Our findings suggest that in humans, learning about reward probability is accompanied by structural alignment of value representations with neural codes for the abstract concept of magnitude.

DOI: https://doi.org/10.7554/eLife.42816.001

**\*For correspondence:**
fabrice.luyckx@psy.ox.ac.uk (FL);
christopher.summerfield@psy.ox.ac.uk (CS)

†These authors contributed equally to this work

**Competing interests:** The authors declare that no competing interests exist.

## Introduction

The ability to learn rapidly from limited data is a key ingredient of human intelligence. For example, on moving to a new city, you will rapidly discover which restaurants offer good food and which neighbors provide enjoyable company. Current models of learning propose that appetitive actions toward novel stimuli are learned *tabula rasa* via reinforcement (*Sutton and Barto, 2018*), and these models explain the amplitude of neural signals in diverse brain regions during reward-guided choices in humans and other animals (*Dolan and Dayan, 2013*; *O'Doherty et al., 2003*; *Schultz et al., 1997*). However, reinforcement learning models learn only gradually, and even when coupled with powerful function approximation methods, exhibit limited generalization beyond their training domain (*Mnih et al., 2015*). This has led to the suggestion they are ill-equipped to fully describe human learning (*Lake et al., 2017*).

By contrast, cognitive scientists have ascribed human intelligence to the formation of abstract knowledge representations (or 'concepts') that delimit the structural forms that new data is likely to take (*Gentner, 2010*; *Kemp and Tenenbaum, 2008*; *Tenenbaum et al., 2011*). Indeed, real-world data can often be described by simple relational structures, such as a tree, a grid or a ring (*Tenenbaum et al., 2011*). Humans may infer relational structure through probabilistic computation (*Kemp et al., 2010*) and a long-standing theory proposes that humans understand new domains by their alignment with existing relational structures (*Gentner, 1983*). However, these models are often criticized for failing to specify how concepts might be plausibly encoded or computed in neural circuits (*McClelland et al., 2010*). A pressing concern, thus, is to provide a mechanistic account of how relational knowledge is encoded and generalized in the human brain (*Tervo et al., 2016*).

The current project was inspired by recent observations that the representational geometry of human neural signals evoked by symbolic numbers respects their relative cardinality. In scalp M/EEG

**eLife digest** Many things in the world have a certain structure to them, which we can use to organize our thinking. To mentally represent your family, for example, you could group your family members into men and women, or group them based on where they live. But a more intuitive approach for most people is to organize family members by generation: child, sibling, parent, grandparent. It is as though we instinctively place each family member along a mental line, from young to old.

We use mental lines to organize other types of information too, most notably numbers. But can we also use them to represent new information? Luyckx et al. trained healthy volunteers to associate pictures of six different colored donkeys with six different reward probabilities. One donkey was followed by reward 5% of the time, another was followed by reward 95% of the time, and so on. Through trial and error, the volunteers learned to rank the donkeys in terms of how likely they were to precede a reward. Luyckx et al. then compared the volunteers' brain activity while viewing the donkeys to their brain activity while viewing the numbers 1 to 6.

The donkeys evoked patterns of electrical brain activity corresponding to the number that signaled their place on a mental line. Thus, donkey 1, with the lowest reward probability, produced a pattern of brain activity similar to that of the number 1, and so on for the others. This suggests that rather than learning in an unstructured way, we use past knowledge of relations among stimuli to organize new information. This phenomenon is called structural alignment.

The results of Luyckx et al. provide the first evidence from brain activity to support structural alignment. They suggest that we use a general understanding of how the world is structured to learn new things. This could be relevant to both education and artificial intelligence. People, and computers, may learn more effectively if taught about the relations between items, rather than just the items in isolation.

DOI: https://doi.org/10.7554/eLife.42816.002

signals, neural patterns evoked by Arabic digits vary continuously with numerical distance, such that multivariate signals for '3' are more similar to those for '4' than '5'. (*Spitzer et al., 2017*; *Teichmann et al., 2018*). In scalp M/EEG signals, neural patterns evoked by Arabic digits vary continuously with numerical distance, such that multivariate signals for '3' are more similar to those for '4' than '5'. Number is a symbolic system that expresses magnitude in abstract form (*Bueti and Walsh, 2009*; *Fischer and Shaki, 2018*; *Walsh, 2003*) and so we reasoned that continuously varying neural signals evoked by numbers might be indexing a conceptual basis set that supports one-dimensional encoding of novel stimuli. In the domain of reward-guided learning, a compact description of the stimulus space projects data into a single dimension that runs from 'bad' to 'good'. Here, thus, we asked humans to learn the reward probabilities associated with novel, high-dimensional visual images, and measured whether the stimuli come to elicit neural patterns that map onto one-dimensional neural codes for numerical magnitude.

## Results

Whilst undergoing scalp EEG recordings, human participants (n = 46) completed two tasks: a numerical decision task and a probabilistic reward-guided learning task (*Figure 1A*, see Materials and methods). In the numerical task, participants viewed rapid streams of 10 Arabic digits (1 to 6) and reported whether numbers in orange or blue font had the higher (Experiment 1a, n = 22) or lower (Experiment 1b, n = 24) average. The reward-learning task was based on the multi-armed bandit paradigm that has been used ubiquitously to study value-guided decision-making (*Dolan and Dayan, 2013*). Participants learned the reward probabilities associated with six unique novel images (colored donkeys), which paid out a fixed reward with a stationary probability (range 0.05–0.95). These probability values were never signaled to the participant but instead acquired by trial and error in an initial learning phase. In the test phase, we asked participants to decide between two successive donkeys to obtain a reward, and estimated trial-wise subjective probability estimates for each bandit by fitting a delta rule model to choices (*Sutton and Barto, 2018*). Throughout these phases, the bandits were never associated with numbers in any way.

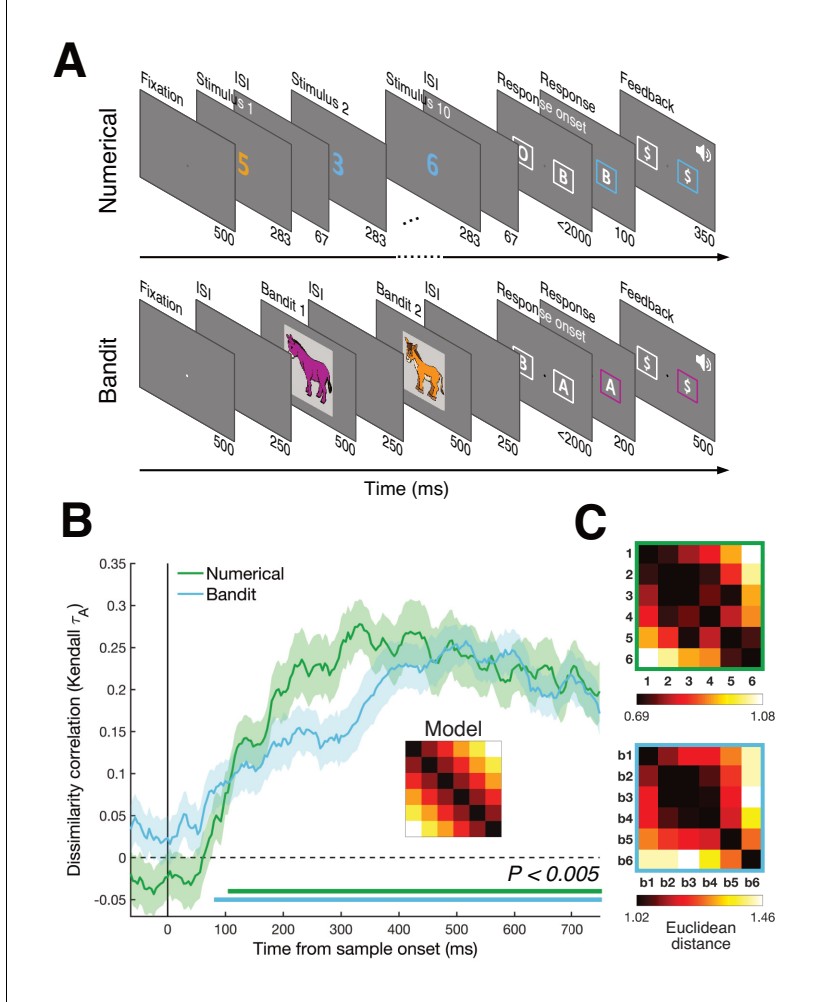

**Figure 1.** Task design and RSA results. (**A**) Humans performed two tasks during a single EEG recording session. In the numerical decision task, participants viewed a stream of ten digits between 1 and 6, deciding whether the blue or orange numbers had the highest/lowest average. In the bandit task, participants learned about the reward probabilities of six images (bandits) and were asked to choose between two successive bandits to obtain a fixed reward. Numbers below each frame show time duration of the frame in ms. (**B**) RSA revealed a numerical and value distance effect from ~100 ms after stimulus onset (bottom colored lines, $P_{cluster}$ <0.005). Inset shows magnitude model RDM. Shaded area represents SEM. Repeating the same analyses with correlation distance (*Figure 1—figure supplement 1*) or splitting the data per task framing group (*Figure 1—figure supplement 2*) provided highly similar results. *Figure 1—figure supplement 3* shows the patterns do not differ between bandits presented first or second. (**C**) Averaged neural RDMs between 200–700 ms for the numerical (top) and bandit (bottom) task show a clear correspondence with the magnitude model.
DOI: https://doi.org/10.7554/eLife.42816.003

The following figure supplements are available for figure 1:

**Figure supplement 1.** RSA results with Pearson correlation distance.
DOI: https://doi.org/10.7554/eLife.42816.004

**Figure supplement 2.** RSA results split per task framing.
DOI: https://doi.org/10.7554/eLife.42816.005

**Figure supplement 3.** RSA results split for first and second bandit.
DOI: https://doi.org/10.7554/eLife.42816.006

# Shared magnitude representation for numbers and probabilistic rewards

Using representational similarity analysis (RSA) (*Kriegeskorte and Kievit, 2013*), we replicated the previous finding (*Spitzer et al., 2017*; *Teichmann et al., 2018*) that patterns of neural activity across the scalp from ~100 ms onwards were increasingly dissimilar for numbers with more divergent magnitude, that is codes for '3' and '5' were more dissimilar than those for '3' and '4' (*Figure 1B*, green line). This occurred irrespective of task framing (report higher vs. lower average) and category (orange vs. blue numbers), suggesting that neural signals encoded an abstract representation of magnitude and not solely a decision-related quantity such as choice certainty (*Spitzer et al., 2017*). Next, we used RSA to examine the neural patterns evoked by bandits. We found that multivariate EEG signals varied with subjective bandit ranks, with bandits that paid out with nearby probabilities eliciting more similar neural patterns (from ~100 ms onwards; *Figure 1B*, blue line).

Our key question was whether there was a shared neural code for numerical magnitude and reward probability. We found that EEG signals elicited by digit '6' were more similar to those evoked by the most valuable bandit, and digit one predicted the bandit least likely to pay out, with a similar convergence for intermediate numbers and bandits (*Figure 2A*). Cross-validation of neural signals elicited by all numbers (1 to 6) and bandits (inverse ranks 1–6) was stable and reliable from 300 to 650 ms post-stimulus, as demonstrated by cross-temporal RSA (*Figure 2B*). We conducted several control analyses to further explore the nature of this effect. The cross-validation effect was not driven by patterns within a single number/bandit pair, as it remained robust to the removal of any one of the six number/bandit pairs (*Figure 2C*). In particular, although the largest number/most valuable bandit cross-validation appeared more dissimilar to other numbers/bandits, the effect persisted when only numbers/bandits 1–5 were included in the analysis (*Figure 2—figure supplement*

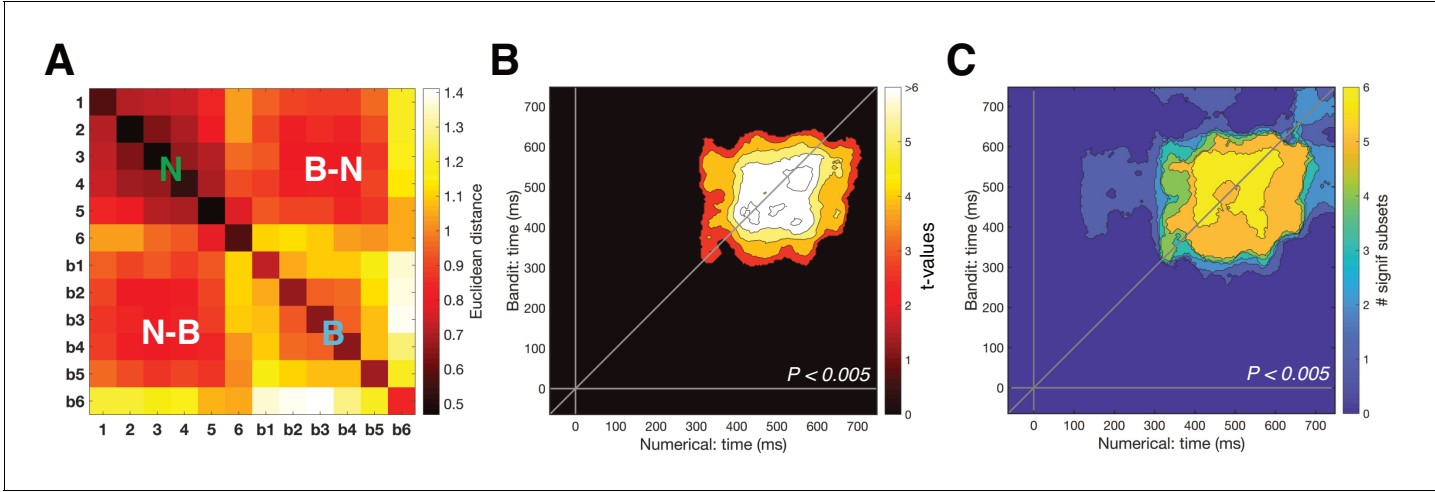

**Figure 2.** Cross-temporal cross-validation RSA. (**A**) Averaged neural RDM from 350 to 600 ms across the numerical (1-6) and bandit (b1–b6) task. Upper left and lower right quadrants show representational dissimilarity for numbers ('N') and bandits ('B'), respectively, that is within-task RDM; lower left/upper right quadrants show cross-validated dissimilarity between numbers and bandits ('N-B'), that is between-task RDM. (**B**) Cross-temporal cross-validated RSA revealed a stable magnitude representation that was shared between the two tasks (P_cluster <0.005) around 350–600 ms. (**C**) To ensure that our cross-validation was not driven by a single number/bandit pair, we systematically removed each number/bandit pair and repeated the cross-temporal cross-validated RSA on the subset data. Each pixel can range from 0 (no significant cross-validation for any number/bandit pair) to 6 (cross-validation always reached significance). Between 400 and 600 ms our cross-validation was robust to the removal of any number/bandit pair (*Figure 2—figure supplement 1* shows the results isolated when excluding number/bandit pair 6).

DOI: https://doi.org/10.7554/eLife.42816.007

The following figure supplements are available for figure 2:

**Figure supplement 1.** Cross-validation after removal of number 6/subjectively highest valued bandit (subset of *Figure 2C*).

DOI: https://doi.org/10.7554/eLife.42816.008

**Figure supplement 2.** We compared the strength of within-subject cross-validation to the between-participants cross-validation, asking whether variations in multivariate neural signals existed that were idiosyncratic within participants.

DOI: https://doi.org/10.7554/eLife.42816.009

*1*). We then asked whether each number was more similar to its equivalent bandit than to other bandits (e.g. number 3 and bandit 3, the 'on-diagonal' information in *Figure 3A*), by computing an 'Exemplar Discriminability Index' (EDI) (*Nili et al., 2016*). Additionally, we asked whether numbers showed a gradually increasing dissimilarity to non-identical bandits (e.g. whether number three was more similar to bandit 2/4 than bandit 1/5, the 'off-diagonal' information in *Figure 3B*). Both of these effects were independently reliable, suggesting not only that each number shares a representation with its corresponding bandit, but that the transitive patterns of encoding numbers and bandits are in a common register in neural signals.

## Relating numerical magnitude representation to choice behavior

Past work has identified overlapping choice biases in numerical cognition and economic decisions (*Kanayet et al., 2014*; *Schley and Peters, 2014*). Next thus, we asked how patterns of behavior in the numerical and bandit tasks were related to one another by creating choice matrices encoding the difference in relative weight given to each number or bandit in the choices made by participants. For the numerical task, we computed decision weights for each number in the choice using an averaging approach (see Materials and methods) and plotted the relative difference in these weights for each combination of numbers (*Figure 4A*). For the bandit task, this was simply the probability of choosing the subjectively highest valued bandit for each combination of bandits (*Figure 4B*). Although choice matrices for numbers and bandits were on average correlated across participants ($\overline{r_\tau}$= 0.32, Z = 5.65, p < 0.0001, Wilcoxon signed rank test), this correlation disappeared when subtracting the group average choice matrices and correlating the residual matrices ($\overline{r_\tau}$= -0.02, Z = -0.49, p = 0.63). In other words, we were not able to identify shared variation in individual weighting of numbers 1-6 and bandits 1-6 in behavior alone, perhaps because of the different nature of the decision required in each task.

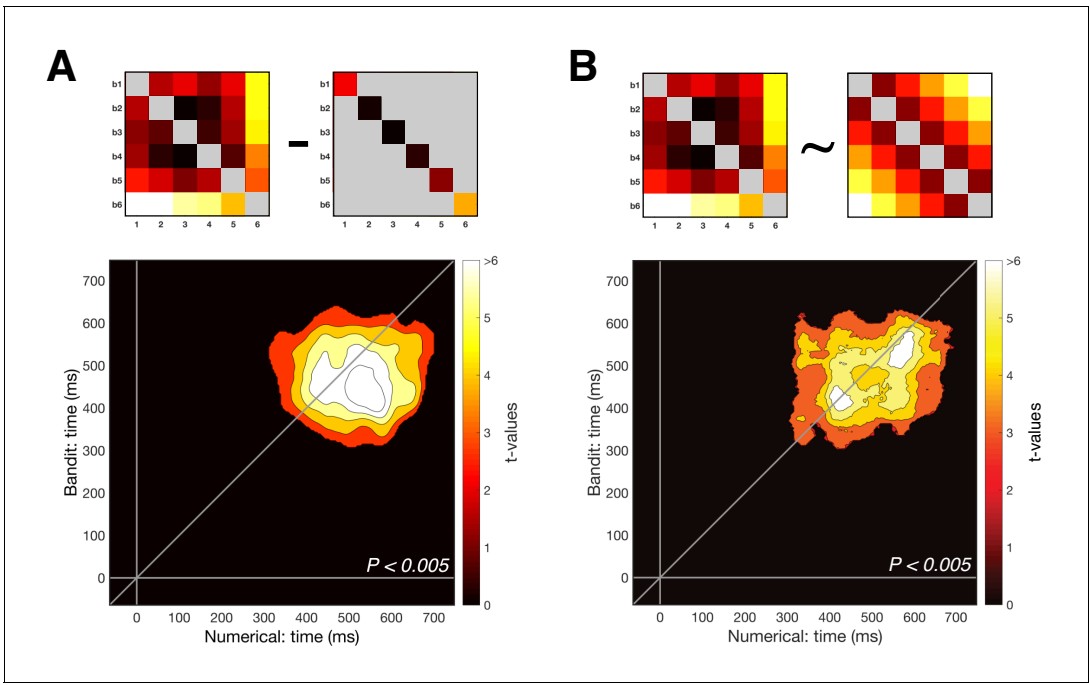

**Figure 3.** Detailed examination of the magnitude pattern in cross-validation. (**A**) The presence of a one-to-one mapping along the number line (i. e. number one to least valued bandit) was tested using the 'Exemplar Discriminability Index' (EDI), a measure that indicates how much better items on average map onto iterations of the same item compared to different items. The EDI is calculated by subtracting the mean on-diagonal distances from the mean off-diagonal distances (top illustration). Significant exemplar discriminability, where numbers most closely resembled their equivalent bandit, arose at the same time as our main findings. (**B**) Removing the diagonal from the lower rectangle in the cross-validation RDM (i.e. the dissimilarity between distributed responses to corresponding stimuli in the two tasks) reproduced the results from our main analysis, suggesting the effect was not mainly driven by matching stimulus pairs (e.g. number six and most valuable bandit), but by a gradual distance effect ($P_{cluster}$ <0.005).
DOI: https://doi.org/10.7554/eLife.42816.010

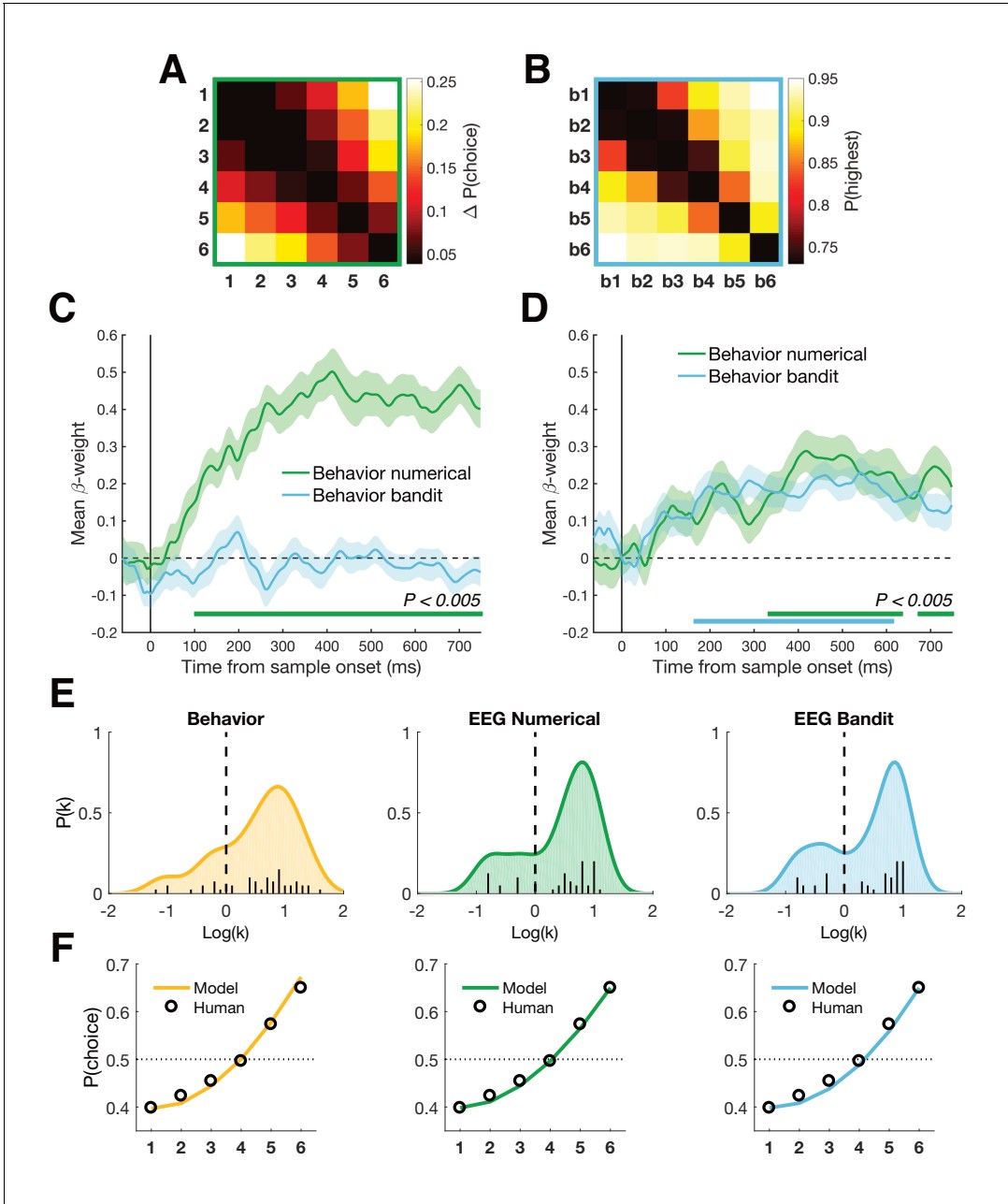

**Figure 4.** Behavioral analyses. (**A**) Group-averaged difference RDM of decision weights in the numerical task. We calculated the weight of each number on participant's choice, independent of color, and created a choice RDM through pairwise differences of weights. (**B**) Group-averaged choice RDM in the bandit task. Each cell contains the probability of choosing the highest valued bandit on trials where bandit *x* and bandit *y* were presented together. Participants never encountered trials where the same bandit was presented twice. (**C**) Both choice matrices were inserted in a multiple regression explaining EEG patterns to establish a potential link between behavior and neural patterns in either task. Only the numerical choice RDM explained neural patterns of the numerical task. (**D**) In contrast, both choice RDMs significantly explained neural patterns in the bandit task, possibly indicating that participants relied on a more general understanding of numbers in both tasks (bottom colored lines, $P_{cluster} < 0.005$). Shaded area represents SEM. (**E**) Distributions of $log(k)$ after fitting a power-law model of the form $x^k$ to the decision weights in the numerical task (yellow) and the average neural patterns in the numerical (green) and bandit task (blue) between 350 and 600 ms (see Materials and methods). In all cases, the best fitting parameter $log(k)$ was significantly greater than 0 (linear), indicating an overweighting/increasing dissimilarity for larger quantities (numbers 5-6 or highest valued bandits). (**F**) Decision weights under the median estimated $k$ for psychometric (yellow) and neurometric (green/blue) fits, compared to the median true human decision weights in the numerical task.
DOI: https://doi.org/10.7554/eLife.42816.011

Subsequently, we asked how these choice matrices explained variance in patterns of neural similarity, that is whether the behavior explained shared variation in the neural structure alignment for numbers and bandits. To this end, we substituted the linear representational distance matrix used for earlier analyses (i.e. one that assumes equal spacing among adjacent numbers and bandits) with the subject-specific choice matrices computed from behavior and repeated the analyses above. This allowed us to ask how patterns of neural similarity among both numbers and bandits were explained by variance in subject-specific choice matrices. For each neural pattern, we used a regression-based approach in which choice matrices from numbers and bandits were entered as competing regressors. Interestingly, we found that choice patterns from the numerical task explained variance in the neural patterns for both numbers and bandits, but choice patterns from the bandit task only explained variance in the neural patterns of the bandit task (*Figure 4C–D*) ($P_{cluster}$ <0.005). One interpretation of this finding is that humans used their intrinsic sense of magnitude when forming neural representations of the bandits, but not vice versa. We note in passing that this asymmetry is not secondary to the ordering of the two tasks, which was fully counterbalanced across participants.

In previous work, we observed that participants tended to give relatively greater weight to larger magnitudes during the numerical decision task, for example numbers '5' and '6' had disproportionate impact on averaging judgments (*Spitzer et al., 2017*). This finding was replicated in the current data (*Figure 4E-F*). Human choices were best fit by a power-law model in which participants averaged and compared distorted numerical values $x^k$ with $k$= 2.04 ± 1.11 ($k$> 1: t(45) = 12.47, p < 0.001). This prompted us to ask whether any shared variance between behavioral choice matrices and neural signals for the two tasks could be explained by subject-specific differences in the pattern of compression or anti-compression in the mental number line, as characterized by this model. Turning to the neural data, we thus generated candidate representational dissimilarity matrices (RDMs) under the assumption that distance in neural space can likewise be non-linear and best described by a distortion given by the same power-law model, that is of the form $x^k$. We found that in both numerical and bandit tasks the best fitting RDM was parameterized by $k$> 1 [numerical: $k$= 1.73 ± 0.82, t(45) = 14.34, p < 0.001; bandit: $k$= 1.72 ± 0.85, t(45) = 13.65, p < 0.001]. In other words, we observed that the anti-compressed number line estimated from behavior was reflected, on average, in both the neural representation of numbers and bandits. However, when we correlated the subject-specific model parameter $k$ from behavior with estimates obtained from the neural data from either task, we found that although the degree of behavioral anti-compression strongly predicted the neural anti-compression for the numerical task ($\rho$= 0.57, p = 0.0004), it did not for the bandit task ($\rho$= -0.13, p = 0.39). One interpretation of this result is that the variance linking the mental number line to the representation of bandits (i.e. from *Figure 4D*) is not simply due to individual differences in compression or anti-compression of the mental representations of numbers and bandits but must lie in a subspace not captured by this simple unidimensional model.

## Dimensionality of magnitude representation

Recent work has suggested that during categorization, posterior parietal neurons in the monkeys are strikingly low-dimensional, as if the parietal cortex were engaging in a gain control process that projected stimulus features or timings on a single axis (*Fitzgerald et al., 2013*; *Ganguli et al., 2008*; *Platt and Glimcher, 1999*; *Wang et al., 2018*). Indeed, we observed a centro-parietal positivity (CPP) that varied with the magnitude of both numbers and reward probabilities (*Figure 5A–B*). This signal resembles a previously described EEG signal, that has been found to scale with the choice certainty in perceptual (*O'Connell et al., 2012*) and economic tasks (*Pisauro et al., 2017*). However, in our numerical task the CPP followed an approximately ascending pattern from lower to higher numbers regardless of task framing (*Figure 5—figure supplement 1*) or color category. The cross-validation effect persisted even after the CPP had been regressed out of the data (*Figure 5—figure supplement 2*). This suggests that (a) the CPP in our task may represent a notion of magnitude, not a certainty signal alone; and (b) that this signal is not the sole driver of our multivariate findings.

Nevertheless, to understand the dimensionality of the number and bandit representations (and the subspace in which they aligned), we used two dimensionality reduction techniques, singular value decomposition (SVD) and multidimensional scaling (MDS). First, using SVD, we systematically removed dimensions from the EEG data and recomputed our number-bandit cross-validation scores (*Figure 5C*). We found that probabilistic reward learning was supported by a low-dimensional neural

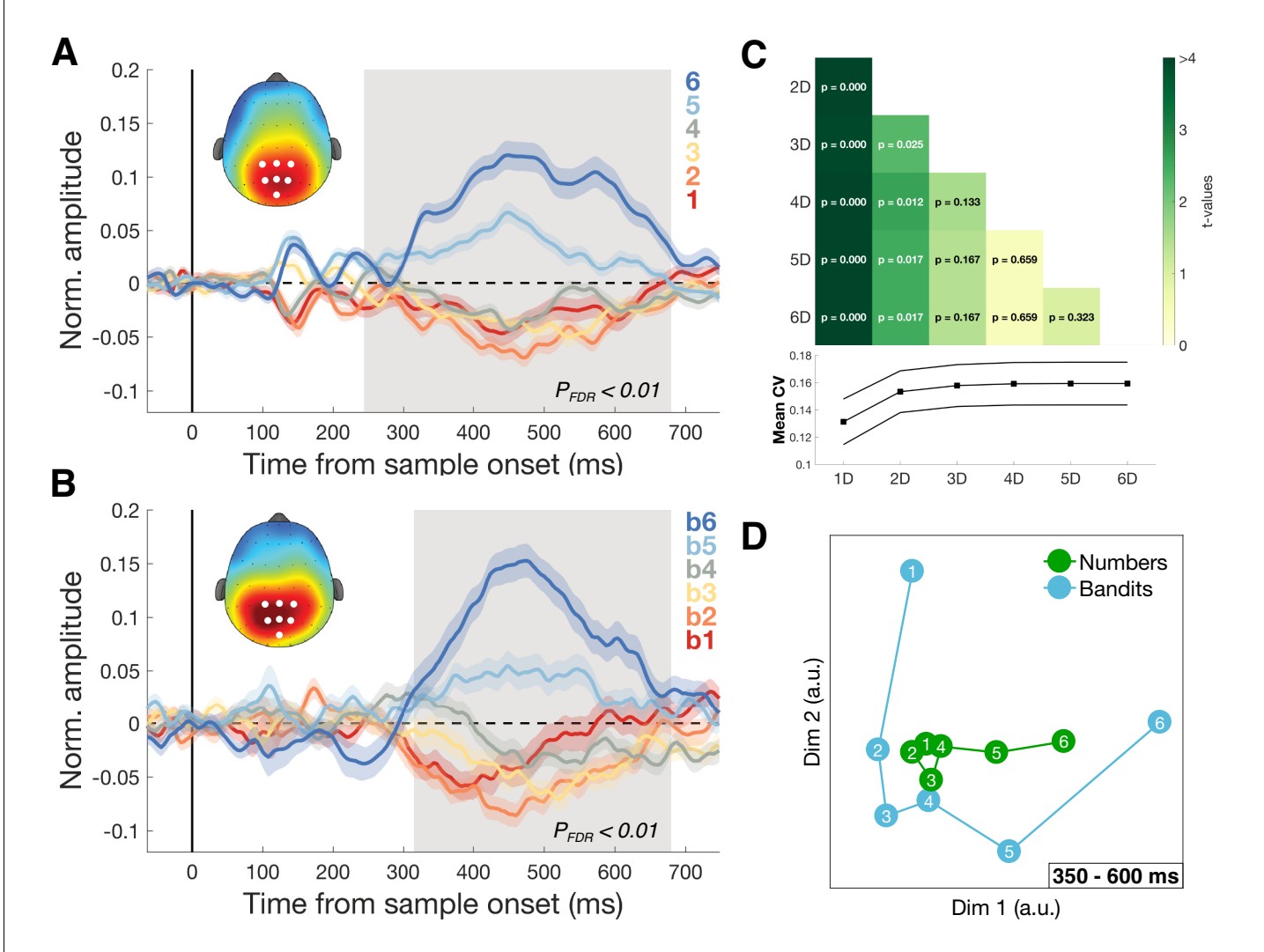

**Figure 5.** Dimensionality of magnitude representation. (**A**) Average normalized amplitudes associated with numbers 1–6, independent of task framing (report highest/lowest) or category (blue/orange) at highlighted centro-parietal electrodes. Grey shaded area shows time of greatest disparity between signals (Kruskal-Wallis, $P_{FDR}$ <0.01). Scalp map inset shows response amplitude for number six during identified time window. Colored shading represents SEM. The ascending direction of the univariate responses was independent of task framing (*Figure 5—figure supplement 1*) (**B**) Equivalent analysis for bandits b1 (lowest value) to b6 (highest value) in the bandit task. Scalp map shows response amplitude for highest subjectively valued bandit b6. (**C**) Dimensionality of the data was iteratively reduced using SVD and the strength of cross-validation under each new dimensionality was assessed by comparing the mean cross-validation in the 350–600 ms time window (bottom plot). Each cell in the grid contains the t- and p-value of a pairwise comparison of mean CV under different dimensionalities of the data. Reduction to one (and to a lesser degree two) dimension(s) significantly reduced the size of the effect. (**D**) Multidimensional scaling (MDS) revealed two principal axes that describe the data: a magnitude axis approximately following the number/bandit order and a certainty axis distinguishing inlying (e.g. 3,4) from outlying (e.g. 1,6) numbers or bandits.
DOI: https://doi.org/10.7554/eLife.42816.012

The following figure supplements are available for figure 5:

**Figure supplement 1.** Univariate centro-parietal signals separated by task and task framing.
DOI: https://doi.org/10.7554/eLife.42816.013

**Figure supplement 2.** We tested whether the univariate CPP amplitude modulations could explain our multivariate findings.
DOI: https://doi.org/10.7554/eLife.42816.014

**Figure supplement 3.** Cross-validation after excluding the first principal dimension.
DOI: https://doi.org/10.7554/eLife.42816.015

magnitude code, with reliable effects persisting when all but two eigenvectors were removed from the data but significantly attenuated when only a single dimension was retained in the EEG data. Indeed, 3D and full (high-dimensional) solutions led to statistically equivalent cross-validation, with some attenuation of the effect when two dimensions were retained and a more dramatic decrease with only a single dimension. To further establish that the cross-validation effect was not solely driven by the observed univariate activity, we again used SVD to remove the first dimension and re-computed the cross-validation statistics (*Figure 5—figure supplement 3*). A significant cluster of cross-validation emerged at the same time as the originally observed effect together with a previously unobserved cluster later in time ($P_{cluster}$ <0.05). In summary, aside from a major univariate component to our cross-validation effect, there remains a shared pattern that lies in higher dimensions of the data.

Secondly, we used MDS to visualize the first dimensions of the concatenated number/bandit data. This disclosed an axis pertaining to magnitude and another approximately corresponding to certainty along which, especially for the bandits, the large (or best) and small (or worst) items diverged from the others (*Figure 5D* and *Video 1*). In other words, the numbers and bandits align principally along a single magnitude axis but with an additional contribution from a second factor potentially encoding choice certainty.

## Neural network simulations

What are the potential benefits of the shared coding scheme we observed in neural signals? One possibility is that shared structure can promote generalization, such that new relational structures (i.e. the transitive relations among bandits as a function of their reward probability) are learned faster and more effectively when an existing scaffold (such as a transitive representation of number) has been previously learned. We are unable to test for this benefit directly in our human data, because all participants were numerate adult humans, denying us an appropriate control condition. However, to demonstrate the theoretical benefit of shared coding at the mechanistic level, we turned to a simple computational tool, a feedforward neural network (*Figure 6A*). Neural networks are not constrained to make inferences over structure, but structured representations may emerge naturally in the weights during training (*McClelland et al., 2010*). Here, we confronted the network with two diffeent stimulus sets in turn that (like our numbers and bandits) shared the same similarity structure. We then asked if the shared structure facilitates retraining on the second set after learning the first. The network was first trained on inputs $x_a$ arriving at input units $X_A$, and after convergence, retrained on inputs $x_b$ fed into units $X_B$ (where $X_A$ and $X_B$ are separate input modules that project to a common hidden layer H). Inputs $x_b$ were 6 random vectors constructed to have the same continuously varying similarity structure as the bandits, whereas inputs $x_a$ consisted of either a different set of six random vectors with the same second-order structure, or a shuffled control lacking the second-order structure. Relearning on $x_b$ proceeded faster when inputs shared a common structure with $x_a$ (*Figure 6B-D*). In a second control, we shuffled the weights $W2$ connecting the hidden layer to the output layer after convergence on inputs $x_a$, destroying the mapping of activity patterns in the hidden layer to the output layer. RSA conducted after retraining revealed reliable cross-validated patterns of activity in the hidden units only for the condition where $x_a$ and $x_b$ shared an underlying structure and weights $W2$ were kept intact (*Figure 6E*), mirroring the result from the human neural data.

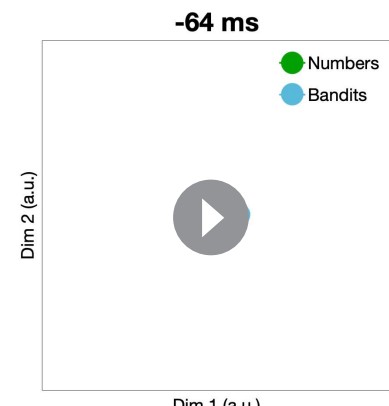

**-64 ms**

- Numbers
- Bandits

Dim 2 (a.u.)

Dim 1 (a.u.)

**Video 1.** The neural geometry of the first two dimensions of both tasks in shared space over time, identified through multidimensional scaling (MSD).
DOI: https://doi.org/10.7554/eLife.42816.016

## Discussion

We report that during a probabilistic reward-guided learning task involving arbitrary images ('bandits'), stimuli with high payout probability

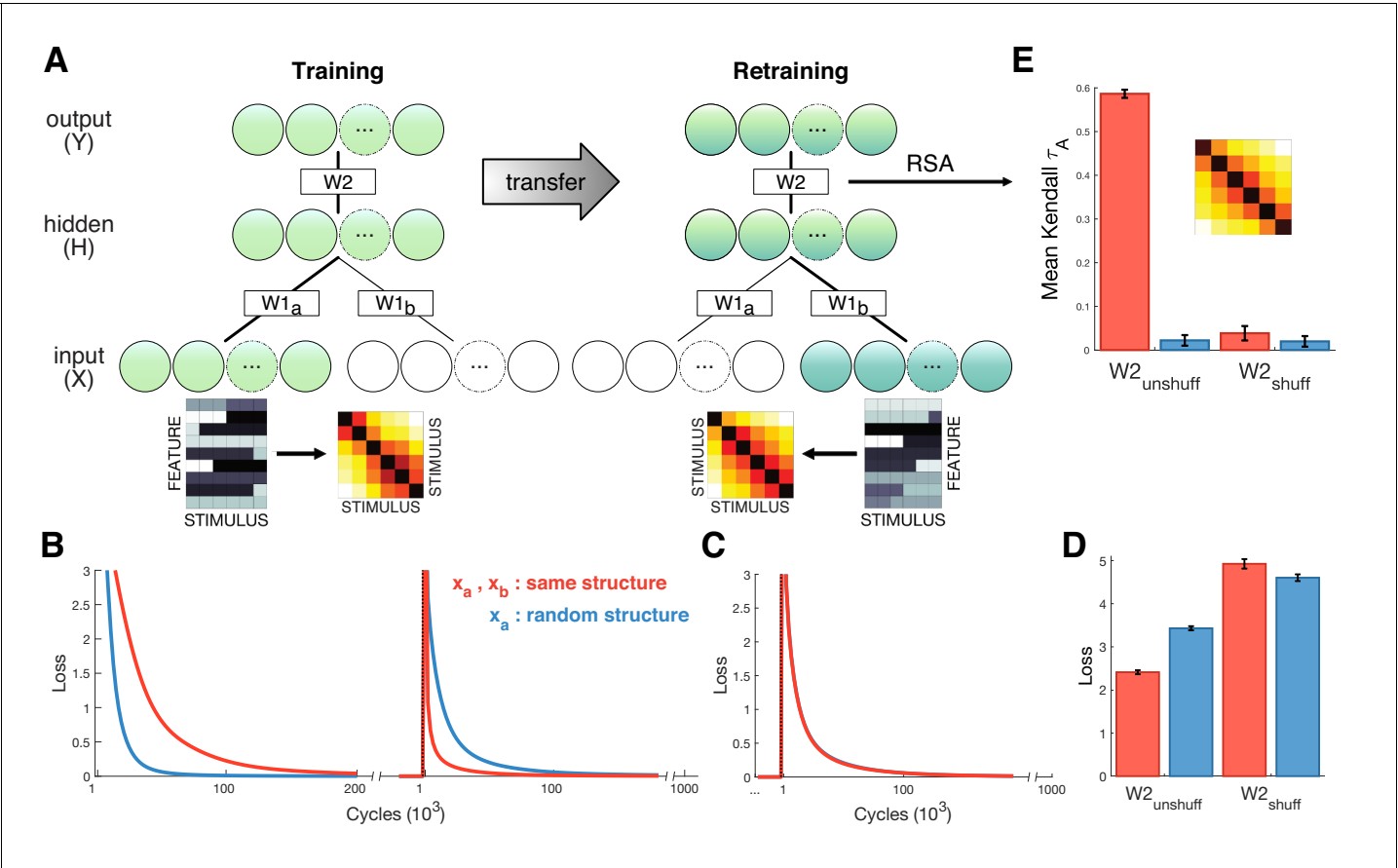

**Figure 6.** Neural network simulations. (**A**) Schematic depiction of network structure and training. The network was first trained to classify inputs $x_a$ fed into units $X_A$ (lower green circles). Inputs $x_a$ consisted of six stimuli that either exhibited gradual increasing dissimilarity or were shuffled as a control (stimulus RDMs shown next to examples of $x_a$ and $x_b$). After convergence, the model was trained on new input $x_b$ that were fed into a separate input stream $X_B$ (lower blue circles). Inputs $x_b$ were different to $x_a$ but exhibited the same similarity structure. (**B**) Loss plotted over the course of training (left panel) and retraining (right panel) for the test (red) and shuffled control (blue) conditions. Learning was faster during training for control stimuli, but retraining was faster when $x_a$ and $x_b$ exhibited shared similarity structure. (**C**) Loss for control simulations where hidden-to-output weights W2 were shuffled between training and retraining, suggesting successful transfer depends on structure encoded in W2. (**D**) Mean loss for first 1000 cycles after retraining. (**E**) Cross-validation RSA on hidden unit activation for all stimuli in $x_a$ and $x_b$ after retraining. Hidden unit activations exhibit shared similarity structure only when W2 remains unshuffled and $x_a$ and $x_b$ share structure. Inset shows cross-validation RDM from hidden units for the test condition. Error bars show SEM over 100 network simulations with different initialization.

DOI: https://doi.org/10.7554/eLife.42816.017

shared a neural code with larger numbers, and those with lower value shared a neural code with lower numbers. We interpret these data as indicating that an abstract neural code for magnitude forms a conceptual basis set or 'scaffold' for learning new information, such as the reward probabilities associated with novel stimuli. Rather than encoding stimulus value in an unstructured value function or lookup table (as is common in RL models), our data suggest that humans project available stimuli onto a low-dimensional axis that runs from 'bad' to 'good'. This neural axis is aligned with the mental number line, suggesting that humans recycle an abstract concept of magnitude to encode reward probabilities.

Learning a structured representation of value will have the benefit of allowing new inductive inferences, such as inferring transitive preferences among economic goods without exhaustive pairwise comparison (*Alfred et al., 2018*), and facilitate read-out in downstream brain areas, related perhaps to the notion of a 'common currency' for reward (*Levy and Glimcher, 2012*).

Our neural network simulations provide a demonstration of how shared structure can promote generalization and thus faster learning, even for stimuli with different physical input features. However, our design was not suited to directly test a benefit of learning between tasks, since our

participants were numerate adults when they entered the experiment. In further work, it would be of particular interest to teach participants two new transitive structures, each associated with a different stimulus set, and test whether the extent to which the neural codes align predicts learning rates for the second stimulus set – a direct prediction that emerges from our computational simulations.

We did, however, find a link between the transitive neural codes and participants' behavior. Choice patterns in the two tasks were positively correlated, and how participants treated numbers in the numerical task was reflected in the neural patterns of both tasks. Furthermore, we found that participants tended to overweight larger numbers and this non-linear weighting correlated highly with non-linear representations at the neural level, at least in the numerical task. Why we did not find a correlation with non-linearities in the neural patterns of the bandit task is unclear. One speculation is that the non-linearity arises at a later stage in the processing of stimuli and is specific to the task at hand.

One major limitation of our approach is the limited spatial resolution of EEG. This leaves open the question of the true dimensionality of the shared neural code. Our investigations using dimensionality reduction techniques indicate that the code in EEG signals is low-dimensional but not simply univariate. Thus, whilst our work is consistent with previous studies showing that the amplitude of centro-parietal EEG signals scales with number (*Spitzer et al., 2017*) and the value of economic prospects, such as food items (*Pisauro et al., 2017*), it also suggests a more complex pattern encoding the shared structure among stimuli defined by transitive relations. However, it would be instructive to measure the effect using techniques that potentially afford higher spatial resolution, such as MEG or fMRI. More generally, however, our work is consistent with theories that have suggested that neural signals for magnitude in the parietal cortex may provide a conceptual bridge between different metrics such as space, time and number (*Bueti and Walsh, 2009*; *Chafee, 2013*; *Fitzgerald et al., 2013*; *Parkinson et al., 2014*; *Walsh, 2003*).

## Materials and methods

### Participants

Forty-nine healthy participants (Experiment 1a = 24, Experiment 1b = 25) participated in behavioral and EEG testing at the University of Oxford. Sample size for Experiment 1a was determined based on common sample sizes in the field and a similar size was used in Experiment 1b for the replication. Two participants from Experiment 1a were excluded from all analyses due to failure to learn in the bandit task (chance level performance) and one participant from Experiment 1b due to excessive movement artefacts in the recorded EEG data. All analyses were performed on the remaining 46 participants (n female = 24, n right-handed = 43, age = 24.7 ± 4.5). All participants had normal or corrected-to-normal vision, with no history of neurological or psychiatric illness. Participants were compensated for their time at a rate of £10/hr plus additional bonuses based on their performance (max. £2.50 in the numerical task and £5 in the bandit task). Informed consent was given before the start of the experiment. The study was approved by the Medical Science Inter-Divisional Research Ethics Committee (R49578/RE001).

### Experimental procedure

Both tasks (numerical and bandit) were run within a single recording session, with the order of tasks counterbalanced between participants. Stimuli were created and presented using the Psychophysics Toolbox-3 (*Brainard, 1997*; *Kleiner et al., 2007*) for Matlab (MathWorks) and additional custom scripts. The tasks were presented on a 20'' screen with a resolution of 1600 × 900, at a refresh rate of 60 Hz and on a grey background. Viewing distance was fixed at approximately 62 cm. The F and J key on a standard QWERTY keyboard served as response keys for left- and right-hand responses, respectively.

#### Numerical task

A trial started with a central dark grey fixation dot lasting 500 ms, followed by 10 Arabic digits at a rate of ~3 Hz (each digit was shown for 283 ms per stimulus with an inter-stimulus interval (ISI) of 67 ms). Numbers were drawn uniformly random from 1 to 6, with half of the stimuli randomly colored in orange and the other half in blue. Sequence generation was unconstrained, except that the blue/

orange means could not be identical. After sequence offset, participants could respond by choosing one of two response boxes on screen containing the options 'O' (orange) and 'B' (blue). Location of response options (left or right box) was fixed within the task but counterbalanced between participants. Left- and right-hand responses were used to select the left and right boxes respectively. When a response was given, the corresponding box would change to the chosen color 100 ms before feedback. In the case of a correct answer, both letters within the boxes were replaced by a dollar sign ('$') accompanied by a high-pitch tone for 350 ms. Conversely, if the response was incorrect or no response was given within 2 s, the boxes would show a dash ('/') and a low-pitch tone was played. The chosen response box remained colored during feedback. The next trial started after an inter-trial interval (ITI) sampled uniformly between 500 and 1500 ms.

Task framing was inverted after testing the first group of participants. In Experiment 1a, participants chose the color associated with the highest average, while in Experiment 1b they chose the color with the lowest average. After 10 (Experiment 1a) or 20 (Experiment 1b) practice trials (excluded from analysis) each participant performed 300 trials in 6 blocks of 50 trials. Participants could take a self-timed break in between blocks.

## Bandit task

Bandits were represented by six unique colored drawings of donkeys, freely available on the internet, and each donkey was colored differently (green, purple, orange, yellow, red and blue) using GIMP (https://www.gimp.org) and superimposed on a light grey background. Each image was associated with one of six stationary reward probabilities, linearly spaced between 0.05 and 0.95 [0.05, 0.23, 0.41, 0.59, 0.77, 0.95], and these were assigned randomly to donkeys for each participant.

The experiment consisted of two learning phases and one test phase. In the first learning phase (L1), all six bandits were presented on screen simultaneously, in a 2 × 3 configuration. Participants could click on each bandit 36 times, in any order they preferred. Every time a bandit was selected, the image was replaced by a feedback sign and the surrounding frame adopted the color of the chosen bandit. Feedback in a successful trial consisted of a centrally presented dollar sign and a high-pitch tone, otherwise a dash appeared paired with a low-pitch tone. Feedback was not drawn probabilistically in this phase: the number of successful trials was determined based on the maximum number of clicks per bandit and its associated reward probability. After a bandit was chosen 36 times, it was masked and remained unavailable until the end of the phase.

In the second learning phase (L2), participants were presented with two random bandits sequentially, identical to the test phase (see below). Phase L2 only differed from the test phase in that it included 50% pseudo-choice trials, where the computer determined which bandit had to be chosen, in order to encourage exploration of all bandits. During these trials, the computer-chosen bandit and corresponding response key were highlighted with a brown frame. All pseudo-choice bandits were assigned equally often and balanced over presentation order. It was emphasized to participants that the pseudo-choice bandits were selected randomly and did not signal the optimal choice. Phase L2 lasted for 2 blocks of 60 trials with a short break in between blocks.

Each trial in phase L2 and in the test phase started with the presentation of a white fixation dot for 500 ms. The fixation dot would disappear 250 ms before presentation of the first bandit. Two bandits were then presented sequentially, each for 500 ms with an ISI of 250 ms. Afterwards, participants had to choose which bandit they preferred. Two response boxes on screen indicated the choice options: 'A' referring to the first bandit presented and 'B' to the second. The location of 'A' and 'B' (left or right box) alternated randomly from trial to trial. The chosen box would then change to the color of the chosen bandit 200 ms before feedback was given. Reward was determined randomly according to the reward probability of the chosen bandit. If the choice was successful, both response boxes would contain a dollar sign and a high-pitch tone played for 500 ms. If the choice entailed no reward or no response was given within 2 s, both boxes would show a dash and a low-pitch tone was played. The fixation dot also turned red when no response was given. No feedback was provided for the unchosen option. A new trial started after an ITI of 500 ms. The test phase consisted of 10 blocks of 60 trials. By the probabilistic nature of the bandit outcomes, the subjective ranking of the bandits could potentially vary over the course of the experiment. It was therefore emphasized to participants before the test phase that the actual reward probabilities of the bandits would never change.

## EEG acquisition

The EEG signal was recorded using 61 Ag/AgCl sintered surface electrodes (EasyCap, Herrsching, German), a NeuroScan SynAmps RT amplifier, and Curry 7 software (Compumedics NeuroScan, Charlotte, NC). Electrodes were placed according to the extended international 10–20 system, with the right mastoid as recording reference and channel AFz as ground. Additional bipolar electrooculography (EOG) was recorded, with two electrodes placed on either temple for recording horizontal EOG and two electrodes above and below the right eye for vertical EOG. All data was recorded at 1 kHz and low-pass filtered online at 200 Hz. All impedances were kept below 10–15 kΩ during the experiment.

## EEG pre-processing

The data from both tasks were pre-processed following the same pipeline, using functions from the EEGLAB toolbox (*Delorme and Makeig, 2004*) for Matlab and custom scripts. First the data were down-sampled to 250 Hz, low-pass filtered at 40 Hz and then high-pass filtered at 1 Hz. The continuous recording was visually screened for excessively noisy channels and these were interpolated by the weighted average of the surrounding electrodes. The data was then offline re-referenced to average reference. In the numerical task, epochs were extracted from 1 s before fixation dot onset to 5.5 s after. In the bandit task, epochs were extracted from 0.5 s before fixation dot onset to 3 s after. Epochs were baselined relative to the full pre-fixation time window. Epochs containing atypical noise (such as muscle activity) were rejected after visual inspection. We then performed Independent Component Analysis (ICA) and removed components related to eye blink activity and other artefacts (manually selected for each participant). Lastly, the trial epochs were split into smaller stimulus epochs for each digit/bandit and re-baselined based on the pre-stimulus onset time window. For the numerical task, these epochs spanned −65 ms to 850 ms relative to stimulus onset. For the bandit task, they spanned −250 ms to 750 ms. Final analyses focused on the overlapping time window of −65 to 750 ms in both tasks.

## Statistical procedure

Experiment 1b was pre-registered as a control experiment (DOI 10.17605/osf.io/ym3gu), directly replicating the bandit task and inverting the task framing for the numerical task. We confirmed our proposed hypothesis that task framing would not affect either the direction of the univariate parietal effects (*Figure 5—figure supplement 1*) or the multivariate patterns (*Figure 1—figure supplement 2*). To increase the power of our analyses, we collapsed the two data sets. All analyses were conducted on the subject level and statistics are reported for the group level.

## Delta-rule model

We estimated subjective probabilities for each bandit using a delta-rule model. On every trial, the model compares the (subjective) values of the two offered bandits and updates the value of the chosen bandit based on the observed reward during the task:

$$V_C(t+1) = V_C(t) + \alpha[R(t) - V_C(t)], \tag{1}$$

where $V_c(t)$ is the value of the bandit chosen by the participant on trial $t$, $\alpha$ the learning rate and $R$ the received reward (either 0 or 1). Value of the chosen bandit is updated for the next trial $t+1$, by taking the difference between the observed reward $R(t)$ and the expected reward $V_C(t)$ modulated by the size of the learning rate. To obtain estimates of the model's choices, the values of the two bandits are passed through a sigmoidal response function:

$$p(A) = \frac{\lambda + (1 - 2\lambda)}{1 + e^{-\Delta V/s}} \tag{2}$$

where $\Delta V = V_A - V_B$, that is the difference in value between the first (A) and second (B) bandit. The policy parameters $s$ and $\lambda$ indicate the slope and termination point (lapse rate) of the logistic choice function. The latter was fixed to 0.05, equivalent to the bounds of the reward probabilities in the task, and the parameters $\alpha$ and $s$ (learning rate and slope) were fit to the data. Best-fitting parameter estimates were obtained by minimizing the negative log-likelihood function using optimization tools in Matlab. Bandit probabilities in the test phase were initialized according to the estimates of

subjective reward probability obtained from free-choice trials from phase L2. Search space was restricted for both parameters between 0.0001 and 0.5. Best fitting parameters were then used to estimate trial-by-trial subjective ranks for all bandits per participant by classifying the bandits according to their subjective values.

## Representational similarity analysis (RSA)

The pre-processed EEG data was first z-scored over all trials, per electrode and time point. To obtain the condition-specific activations at each time point and electrode, we constructed a design matrix using dummy coding for each condition within a task (numbers 1 to 6 and the six bandits) and subsequently estimated beta coefficients using a linear regression model for each number or (ranked) bandit. These beta coefficients then reflected the trial-average response per condition at each time point and electrode. We then calculated the Euclidean (or correlation) distance between the whole-scalp neural signals of each condition pair (e.g. bandit 1 and bandit 2), resulting in a 6 × 6 representational dissimilarity matrix (neural RDM) at each time point for both tasks separately (*Figure 1C*). Neural RDMs were smoothed over time through convolution with a 60 ms uniform kernel. To test for patterns of numerical distance in the neural RDMs, we created a 6 × 6 magnitude model RDM in which the predicted dissimilarity linearly increased from 0 to 1 as a function of the numerical difference between two numbers. The upper triangles of the model and neural RDM were subsequently correlated using Kendall Tau-a rank correlation (*Figure 1B*) (*Nili et al., 2014*).

In the cross-validation analyses, we followed a similar pipeline as the within-task RSA described above, except that beta estimates of condition activation from both tasks were concatenated before calculating the Euclidean distance between all 12 conditions, resulting in a 12 × 12 RDM containing both *within-task* (e.g number-number) and *between-task* (e.g. number-bandit) dissimilarities (*Figure 2A*). To account for potential differences in the time at which a magnitude code was decodable, we conducted this analysis for all possible combinations of time points (cross-temporal RSA) (*King and Dehaene, 2014*). Neural RDMs were smoothed over time through convolution with a 60 x 60 ms uniform kernel. In cross-validation, model RDMs were correlated with the lower rectangle of the neural RDM, containing the between-task dissimilarities. The diagonal of the rectangle was included, since on-diagonal information is non-redundant in cross-validation. Correlations were baseline corrected by subtracting the average correlation in the pre-stimulus period to not bias our cluster-identifying algorithm. Significant clusters were identified using cluster-corrected nonparametric permutation tests (iterations = 1000, cluster-defining and cluster-level thresholds at p<0.005, unless stated otherwise) (*Maris and Oostenveld, 2007*).

We tested to what extent the choice behavior of a task was reflected in the neural patterns through multiple regression (*Figure 4A–D*). For the numerical task, a model RDM was calculated for each participant based on the differences in choice probability for each number pair (see below). The bandit model RDM was constructed by taking the average probability of choosing the most valuable bandit (according to the delta-rule model estimates) for any bandit pair. Both models were vectorized and z-scored and entered as two regressors in a model explaining variance in the neural RDM of either the numerical task (*Figure 4C*) or the bandit task (*Figure 4D*). Similarly, for the CPP control analyses (*Figure 5—figure supplement 2D–E*), a model RDM was constructed based on the differences in average peak activity per condition (see below). The CPP model was then entered in a multiple regression together with the magnitude model, to determine whether the variance in neural patterns could be explained by a magnitude code over and above the univariate findings.

## CPP analysis

Normalized EEG epochs were averaged for each digit (independent of color category) or for each subjective bandit rank. Based on previous research (*Spitzer et al., 2017*; *Twomey et al., 2015*), we selected seven centro-parietal electrodes (CP1, P1, POz, CPz, CP2 and P2) and averaged the event-related potentials (ERP) over these electrodes for each stimulus. Next, we sought to identify for each task the time window where the disparity in ERP signals was greatest between stimulus types, using a non-parametric omnibus test (Kruskal-Wallis test) at every time point. To avoid circular inference, the test was performed in a leave-one-out fashion, determining significance based on the remaining 45 participants. The largest cluster of adjacent significant time points (p<0.01) was then determined using FDR correction for multiple comparisons (*Benjamini and Hochberg, 2009*). Activity within

each individual's time window was then averaged for each condition and differences in the ERP averages were taken to construct RDMs that were used as control models for RSA (*Figure 5—figure supplement 2*).

## Dimensionality reduction

We assessed the dimensionality of the neural data through Singular Value Decomposition (SVD), a method that allows to efficiently obtain principal components in the data through linear transformation. After estimating beta coefficients for each condition, we used SVD at each time point to obtain the diagonal matrix $\Sigma$ that contained the six singular values. We then systematically reduced the dimensionality of the data by removing the last column of $\Sigma$, reconstructed the data under the reduced dimensionality and followed the rest of the cross-validation pipeline as described above. Changes in the strength of the cross-validation were tested by comparing the model – EEG RDM correlations averaged over the 350-600 ms time window, under the different dimensionalities of the data (*Figure 5C*).

## Psychometric model

To estimate numerical magnitude distortion in the numerical task, we fitted a psychometric model to the choice data (*Figure 3F*). The model is adapted from that described in *Spitzer et al. (2017)*. First, input values (numbers 1 – 6 in the task) are normalized between 0 and 1 in six equidistant steps. These normalized values $nX$ are then transformed into a subjective decisional value $\hat{X}$ by exponentiating them with free parameter $k$.

$$X = nX^k \tag{3}$$

When $k = 1$, $\hat{X}$ is equal to $nX$. When $k > 1$, decision values are exponential, giving relatively higher weights to larger numbers and being more indifferent to smaller numbers. Conversely, when $k < 1$, decision values are compressed, giving lower weight to the larger numbers. Next, we model the trial-level decision-value as the sum over all samples $\hat{X}$.

$$DV = \sum_{i=1}^{10} \hat{X}_i \cdot c_i \cdot l^{10-i} \tag{4}$$

where $c_i$ is an indicator variable that codes for stimulus category (*orange* = -1, *blue* = 1) and $l_i$ is a leakage term that exponentially discounts earlier samples in the stream. Finally, the model uses a logistic function for computing choice probabilities:

$$p(\boldsymbol{blue}) = f - \frac{1}{1 + e^{-(B+DV)/s}}, \tag{5}$$

where $p(blue)$ is the probability of choosing the blue category. $s$ is the inverse slope of the logistic choice function and $B$ captures a simple response bias toward one of the two categories. Finally, $f$ is an indicator variable that codes for the framing of the task, inverting the choice probability for the low frame ($f = 1$) compared to the high frame ($f = 0$). Best fitting parameters for $k$[0.01 – 10], $l$[0 – 1], $m$[unconstrained] and $s$ [0.001 – 8] were obtained using standard optimization tools in Matlab.

Model-free decision weights were estimated for each participant using an averaging approach to compare to the model predictions. Choice probabilities for each digit were calculated by averaging over all responses to trials where that digit was present.

## Neurometric mapping

In order to obtain an estimate of the distortion in neural representational geometry, we generated a range of candidate model RDMs computed from features that were distorted by a parameter $k$, analogous to the psychometric model. Six equidistant values between 0 and 1 were raised to the power of $k$ (*Equation 3*) and model RDMs were constructed based on the Euclidean distances after distortion. Parameters that fitted the neural data best were found through exhaustive search through values of $k$ from 0.35 – 3 in steps of 0.01, iteratively computing the Kendall Tau-a correlation between the upper triangle of the model RDM and the neural RDM averaged over a time window between 350 and 600 ms. The distortion $k$ associated with the highest correlation was determined in each

participant individually and subjected to group-level analysis. Representational geometries for the two tasks were stable in this time window based on the analysis depicted in *Figure 2B*.

## Neural network

We constructed a simple feedforward neural network with 21 input units (1 bias unit and two input modules of 20 units, $X_A$ and $X_B$), 10 hidden units $H$ and 10 output units $Y$ (*Figure 6A*). The network was trained (learning rate = 0.001) to map inputs $x_a$ and $x_b$, both consisting of 6 stimuli with 20 features each, onto 6 random vectors. Both inputs were generated by drawing random values from a standard normal distribution. In the crucial test condition, both $x_a$ and $x_b$ shared a second-order structure of gradually increasing dissimilarity. This was achieved by successively flipping the sign of two more features for each adjacent stimulus. The network learned to minimize the cost-function (cross-entropy) with respect to the supervision signal and via backpropagation, first on inputs $x_a$ arriving at input units $X_A$, and after $10^6$ iterations it was then retrained on inputs $x_b$ fed into units $X_B$.

Two control conditions were included to assess the contribution of shared structure to retraining performance. In one control, the model was initially trained on an input $x_a$ that was entirely random, that is not constructed using the sign-flipping method. In the second control, hidden-to-output weights *W2* were shuffled before retraining on $x_b$. Simulations were run 100 times for each of the conditions ($x_a$ and $x_b$ shared structure vs. $x_a$ was random; *W2* unshuffled vs. shuffled).

Next, RSA was performed on the hidden unit activations, after convergence had been achieved on retraining, to test for shared representational structure in the hidden units. Each stimulus from $x_a$ was fed into $X_A$ and $x_b$ into $X_B$ to obtain hidden unit activations for each stimulus. A cross-validation RDM was then constructed based on the differences in activations between $x_a$ and $x_b$ for each stimulus and correlated with a magnitude model RDM (*Figure 6E*).

## Data and code availability

All code and materials to reproduce the analyses and experiments are available at https://github.com/summerfieldlab/Luyckx_etal_2019. (*Luyckx, 2019*; copy archived at https://github.com/elifes-ciences-publications/Luyckx_etal_2019). Data to reproduce the results available from the Dryad Digital Repository: https://doi.org/10.5061/dryad.7k7s800. Raw EEG data files are available upon request.

## Acknowledgements

The authors thank Zeb Kurth-Nelson, Laurence Hunt and Gaia Scerif for their insightful comments and Mark Stokes for providing access to EEG equipment.

## Additional information

### Funding

| Funder | Grant reference number | Author |
| --- | --- | --- |
| European Research Council | Consolidator Grant CQR01290.CQ001 | Christopher Summerfield |
| Deutsche Forschungsgemeinschaft | SP 1510/2-1 | Bernhard Spitzer |

The funders had no role in study design, data collection and interpretation, or the decision to submit the work for publication.

### Author contributions

Fabrice Luyckx, Conceptualization, Data curation, Software, Formal analysis, Validation, Investigation, Visualization, Methodology, Writing—original draft, Writing—review and editing; Hamed Nili, Supervision, Methodology, Writing—review and editing; Bernhard Spitzer, Conceptualization, Software, Supervision, Methodology, Writing—review and editing; Christopher Summerfield, Conceptualization, Resources, Software, Supervision, Funding acquisition, Methodology, Writing—original draft, Project administration, Writing—review and editing

**Author ORCIDs**
Fabrice Luyckx http://orcid.org/0000-0003-0656-538X

**Ethics**
Human subjects: Informed consent was given before the start of the experiment. The study was approved by the Medical Science Inter-Divisional Research Ethics Committee at Oxford University (R49578/RE001).

**Decision letter and Author response**
Decision letter https://doi.org/10.7554/eLife.42816.022
Author response https://doi.org/10.7554/eLife.42816.023

## Additional files

**Supplementary files**
• Transparent reporting form
DOI: https://doi.org/10.7554/eLife.42816.018

**Data availability**
All data necessary to reproduce the results are available on https://dx.doi.org/10.5061/dryad.7k7s800. All code and materials to reproduce the analyses and experiments are available at https://github.com/summerfieldlab/Luyckx_etal_2019 (copy archived at https://github.com/elifesciences-publications/Luyckx_etal_2019).

The following dataset was generated:

| Author(s) | Year | Dataset title | Dataset URL | Database and Identifier |
|-----------|------|---------------|-------------|-------------------------|
| Luyckx F, Nili H, Spitzer B, Summerfield C | 2019 | Data from: Neural structure mapping in human probabilistic reward learning | https://dx.doi.org/10.5061/dryad.7k7s800 | Dryad Digital Repository, 10.5061/dryad.7k7s800 |

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
