## [Decision Letter]

Thank you for submitting your article "Neural structure mapping in human probabilistic reward learning" for consideration by *eLife*. Your article has been reviewed by three peer reviewers, including Daeyeol Lee as the Reviewing Editor and Reviewer #1, and the evaluation has been overseen by Joshua Gold as the Senior Editor. The following individual involved in review of your submission has agreed to reveal their identity: Matthew Chafee (Reviewer #3).

The reviewers have discussed the reviews with one another and the Reviewing Editor has drafted this decision to help you prepare a revised submission.

Summary:

Neural activity recorded with EEG display similar responses associated with two different types of magnitude information when the subjects process a series of rapidly presented numbers and when the subjects learned a set of reward probabilities associated with arbitrary visual. This suggests that the brain uses the dimension of magnitude as a scaffolding to incorporate new information.

Essential revisions:

1) Results do not convincingly demonstrate that the similarity in the neural activity induced by numbers and probability is reflected in the pattern, rather than in a univariate activity.

The authors use representational similarity analysis (RSA), based on a Euclidean distance measure, to compare patterns of activation. Previous research has shown univariate monotonic representations of value in several brain areas, including posterior parietal cortex (e.g. encoding of both magnitude and probability of expected juice, Platt and Glimcher 1999). If I understand correctly, a crucial aspect of the results here is that they hold after accounting for a univariate effect, so that the similarity is indeed in the *pattern* of activation. The univariate effect was estimated based on the mean response of seven centro-parietal electrodes. It is not clear that this is enough to rule out the possibility that the results still reflect a univariate effect – for example, there may be additional univariate effects in other electrodes, which were not accounted for. Why did the authors choose to use a Euclidean distance measure, which by definition will also pick up univariate effects? How about using the Pearson correlation (or 1-Pearson correlation) instead? This measure will not reflect any univariate effects.

2) The main result shown in Figure 1C needs clarifications.

Figure 1C shows the similarity of within-task and cross-task decoding. Although the results from the within-task coding look clean, the results from the cross-task decoding provide only limited support to the shared magnitude representation. Namely, the activity associated with the maximum quantity (6) looks distinct from the remaining 5 values, although it looks more similar to larger values (4 and 5) than the smaller ones. This is also very different from the results from the simulation of network model (Figure 2E). In addition, there is not a strong tendency for EEG feature vectors to be more similar along the diagonal within the upper right and lower left blocks themselves (representing the same magnitudes across the two tasks) versus off the diagonal (representing different magnitudes in the two tasks) within these blocks. It seems as if the neural encoding of individual magnitudes was lost between the two conditions. The authors performed additional tests in Figure 1E by removing individual quantities, but the results show the average of all those control analyses. The authors should test the similarity after removing just the maximum value and see whether the remaining values show cross-task similarity. If not, they should provide some explanation as to why their findings might be entirely driven by the maximum value.

Is it possible that the similarity reflected the brain being in a “magnitude evaluation” state rather than “representing magnitude 6” state between the two tasks (as one example)? If the neural encoding of specific magnitudes does not generalize across tasks, to what extent can it be said that the two tasks share a single underlying “mental number line”? Figure 3A (bottom panel), Figure 2—figure supplement 2, and associated analysis seem to get at this point but the link here was not entirely clear. Figure 3A (bottom panel) shows that removing the diagonal from the cross-validation across tasks did not remove the cross validation between tasks over time (that is, EEG feature vectors at one time point in the bandit task were similar to EEG feature vectors at multiple time points in the number task and vice versa). This result is interpreted to indicate that the cross-temporal similarity did not depend on shared single-quantity coding across tasks (“but by a gradual distance effect”). Could it also reflect a shared state change across tasks related to the cognitive process of quantitative comparison, rather than magnitude encoding as suggested by a shared number line? It could be useful to consider more extensively the limit to which the EEG data indicate that the neural representation of specific magnitudes generalizes across tasks.

Another interesting aspect of the dissimilarity data across tasks is that there seems to be greater EEG vector similarity (“confusion”) off-diagonal in the number task (Figure 1C, upper left quadrant) relative to the bandit task (lower right quadrant). Does this imply different “width” of neural tuning for magnitude in the two tasks? Although it is difficult to extrapolate from single neuron to EEG studies, single neuron recording in prefrontal and parietal cortex of monkeys performing quantity evaluation tasks such as those conducted by A Nieder, EK Miller, and colleagues demonstrated that single prefrontal and parietal neurons have tuning functions for magnitude that are centered on different “preferred” quantities but are broadly tuned. This seems at least conceptually consistent with the similarity between EEG feature vectors seen off-diagonal in the upper left quadrant of Figure 1C. Conversely, similarity between EEG feature vectors seems much more concentrated along the diagonal in the bandit task (lower right quadrant, Figure 1C). This could potentially imply a neural code that is more like a look-up table in the bandit task in which a specific EEG feature vector was only seen for bandit 5 and no other bandit for example). That could suggest two magnitude representations in the two tasks, one tuned for magnitude (number task), one implementing a lookup table for magnitude (bandit task). Some comment about this potential different in magnitude representations could be useful. This would not invalidate the idea of a shared magnitude representation, but it might identify more clearly aspects in which the magnitude representation was modulated across contexts.

3) Behaviors need to be documented more carefully.

The behavioral in both tasks seems very important for the research question, yet the behavior is hardly reported. The actual choices, as a function of the digits/bandits presented are not reported. In the supplementary material, Figure S3 presents some results, but these were not completely clear to me. The authors replicate the distortion in weighting numbers in the numerical task, which they have reported previously, and also provide convincing evidence that this distortion is associated with the neural signal. However, they do not show whether the behavioral distortions in the two tasks are correlated across subjects, and they do show that the neural distortions are not correlated (Figure 3B). I did not follow the logic of Figure 3C – how exactly were choice distortions in the numerical task used? Why were the distortions estimated from the bandit task regressed out? Isn't the idea that the same system is responsible for distortions in both tasks? Or do you assume separate sources for distortions, only some of which are shared across tasks?

The main thrust of the argument was that forming generalized representations of the structure of input data (such as the concept of magnitude) in one context accelerates learning in new contexts. The network model examined this question, but the behavioral data in human subjects was not interrogated to show that learning in the second task was accelerated relative to the first (unless I missed it). It is impressive that distortions in magnitude representation (biased encoding of larger magnitudes) were shared across task contexts, suggesting a shared magnitude representation. However, the behavioral or computational benefits of the underlying generalized representation in terms of accelerating learning in new contexts did not appear to be explicitly addressed. If that is the case it could be useful to consider this limitation (and potentially how it could be addressed in future studies).

4) Some methods need to be described in more details so that the manuscript is more accessible to non-expert readers.

What aspect of neural activity comprised the feature vector entered in the RSA? The Material and methods states “linear regression (GLM) was used to estimate beta coefficients for each number or (ranked) bandit for each electrode and time point.” I was puzzled what data went into the regression model and how the model was specified. Am I right that there were separate regressors for each individual number and bandit, each coded by a dummy variable, and that the regression coefficient associated with each variable (at each electrode and time point) quantified the proportion of variance in z-scored EEG signal amplitude explained by the presence (absence) of that specific quantity over stimulus repetitions (trials)? One critical piece of information missing for me was the source of the variance the model was trying to explain (variability over stimulus presentations?) It would be important in addition to clarify whether, rather than the dummy coding above, the GLM model included numerical quantity and reward (bandit) probability as scalar independent variables (which would test a linear relation between neural activity and magnitude). Some additional explanation about the steps in the analysis, specification of the model, whether the model explained variance over stimulus repetitions (or trials) or some other source, and what the meaning of the resulting beta coefficients was in neural terms would help increase the clarity of the main finding.

---

## [Author Response]

Essential revisions:1) Results do not convincingly demonstrate that the similarity in the neural activity induced by numbers and probability is reflected in the pattern, rather than in a univariate activity.The authors use representational similarity analysis (RSA), based on a Euclidean distance measure, to compare patterns of activation. Previous research has shown univariate monotonic representations of value in several brain areas, including posterior parietal cortex (e.g. encoding of both magnitude and probability of expected juice, Platt and Glimcher 1999). If I understand correctly, a crucial aspect of the results here is that they hold after accounting for a univariate effect, so that the similarity is indeed in the pattern of activation. The univariate effect was estimated based on the mean response of seven centro-parietal electrodes. It is not clear that this is enough to rule out the possibility that the results still reflect a univariate effect – for example, there may be additional univariate effects in other electrodes, which were not accounted for. Why did the authors choose to use a Euclidean distance measure, which by definition will also pick up univariate effects?. How about using the Pearson correlation (or 1-Pearson correlation) instead? This measure will not reflect any univariate effects.

Thanks for this point. The reviewers are right that previous studies have demonstrated univariate encoding of probability and value in the parietal cortex. The Platt and Glimcher study cited of course involves recording from single cells sampled to have response fields at specific locations (i.e. congruent with one of the targets), and thus one might not expect it to translate directly to macroscopic signals measured in human scalp EEG. It is also true that we have previously shown monotonic univariate encoding of number in EEG (Spitzer et al., 2017) and others have shown that the subjective value of economic prospects, such as food items, is encoded in the CPP (Pisauro et al., 2017). Importantly however, our study is the first to demonstrate such signals in human EEG during a probabilistic reward learning task, involving completely arbitrary visual stimuli. This advance notwithstanding, we agree that our empirical results would be less novel if all effects were simply attributable to a common univariate CPP signal (although our theoretical account would still stand). However, our control analysis and in particular the careful consideration of the dimensionality of the signal patterns (SVD analyses) seem to suggest that this is not the case.

The reviewer points out that correlation distance, which is scale invariant and thus removes the univariate effect, would be an appropriate way to test this. We agree. In fact, we tried both measures (Euclidean and correlation distances) during our original analysis and they yielded similar results. We chose to display the Euclidean distance because it is (arguably) a more interpretable measure. However, in response to the reviewers’ suggestions, we have now added a supplementary figure that presents the same results using (Pearson) correlation distance (Figure 1—figure supplement 1). As the reviewer will see, correlation distance yields very similar results to Euclidean distance.

We agree with the reviewers that activity from other electrodes not included in our subset of 7 electrodes could contain residual univariate effects. Since our EEG data is referenced to the mean of all electrodes, the sum of all recorded activity across the scalp is zero. This makes it more challenging to deal with this issue (i.e. one inevitably has to sub-select electrode clusters to plot a univariate effect, or else it will simply average out across the scalp). However, another way to test whether our effects are univariate or multivariate is to remove dimensions from the data, for example using singular value decomposition (SVD), and test whether our effect persists after removal of the first dimension. We have now added a supplementary analysis (Figure 5—figure supplement 3) using this approach and recomputing the cross-validation statistics. If the effect were purely univariate, removing the first dimension would eliminate it in cross-validation. However, with the first dimension removed, still, a significant cluster of cross-validation emerges at the same time as the originally observed effect, and a previously unobserved cluster also emerges later in time. Our best interpretation of these findings aligns with that proposed in the initial manuscript: that there is a major univariate component to our cross-validation effect (as would be expected from previous studies), but that the shared pattern is also observed in higher dimensions of the data.

2) The main result shown in Figure 1C needs clarifications.Figure 1C shows the similarity of within-task and cross-task decoding. Although the results from the within-task coding look clean, the results from the cross-task decoding provide only limited support to the shared magnitude representation. Namely, the activity associated with the maximum quantity (6) looks distinct from the remaining 5 values, although it looks more similar to larger values (4 and 5) than the smaller ones.

Thanks for this. In part, the reviewer is right; the greatest distance is between quantity 6 and the other 5 quantities. We think this is due to the “anti-compression” in the number line in this task, an effect we have reported previously (Spitzer et al., 2017) and replicate here (Figure 4E-F). In part, this effect appears somewhat visually exaggerated due to the scaling of the 12 x 12 cross-validation RDM and nonlinearities in the colormap (the “hot” map in Matlab). However, the reviewer is right to ask whether the effect persists when number 6 is excluded. Indeed, it does. In Figure 2C, we excluded each of the numbers in turn and recalculated the cross-validation effect, plotting for each pixel the number of times for which our cross-validation reached significance (i.e. zero to 6, the maximum number of possible folds of the data). Now, additionally, in Figure 2—figure supplement 1, we show the data specifically excluding number 6. As can be seen, the cross-validation effect is still present, albeit reduced in magnitude as one might expect.

This is also very different from the results from the simulation of network model (Figure 2E).

This is a good point. We could have been clearer that our network simulations were not intended to explicitly simulate the human data from this task, but rather to show (as a proof of concept) that prior learning of data with comparable similarity structure can facilitate relearning. This is because in the simulations, the level of prior experience is under our control (we either expose the network to a different dataset with shared structure, or do not). Clearly, this control is unavailable for the human data, because our participants (healthy adults) are all numerate and thus have substantial prior learning of magnitude structure. Instead, the goal of our simulations was different – to show why learning (or having learned) such a representation might be beneficial for new learning, i.e. to provide a normative motivation for the coding scheme that our data reveal. We now acknowledge these limitations more explicitly both when describing the neural network and in the Discussion.

In newer work building on the present findings, which is ongoing in the lab, we are conducting experiments that resemble far more closely the neural network simulations (i.e. testing for learning and generalisation in wholly novel domains with shared or unshared structure). We hope that this work will be ready for dissemination soon.

In addition, there is not a strong tendency for EEG feature vectors to be more similar along the diagonal within the upper right and lower left blocks themselves (representing the same magnitudes across the two tasks) versus off the diagonal (representing different magnitudes in the two tasks) within these blocks. It seems as if the neural encoding of individual magnitudes was lost between the two conditions.

We agree that, at least visually, it might seem as if on-diagonal elements in the between-task cross-validations (lower left and upper right) do not appear very strong. In part, greater on-diagonal similarities *within* tasks (upper left and lower right) probably follow naturally from the additional boost to decoding given by shared perceptual features (i.e. Arabic numeral 3 is more similar to itself than to any of the donkey images). This exaggerates the scaling of the figure which may obscure more subtle differences in similarity in the between-task cross-validation. To test this explicitly, we examined the on- and off-diagonal information separately. We included new control analyses and a new figure (Figure 3) specifically tackling this question. We describe the ‘Exemplar Discriminability Index’ (EDI; Nili et al., 2016), a measure that indexes the similarity of stimuli of equivalent magnitudes (on-diagonal) compared the other stimuli (off-diagonal) (Figure 3A). In a second control analysis, we excluded the on-diagonal information to probe whether our cross-validation was not solely driven by one-to-one mapping of magnitudes, but also by the pattern of gradually increasing dissimilarity (Figure 3B). Both effects were independently reliable. In other words, number 3 is more similar to bandit 3 than to other bandits (on-diagonal or exemplar cross-validation effect) and number 3 is also more similar to bandits 2 and 4 than to other bandits (off-diagonal distance effect, statistics computed after excluding the diagonal). This seems to support the claims we make in the paper.

The authors performed additional tests in Figure 1E by removing individual quantities, but the results show the average of all those control analyses. The authors should test the similarity after removing just the maximum value and see whether the remaining values show cross-task similarity. If not, they should provide some explanation as to why their findings might be entirely driven by the maximum value.

Thank you – we believe we have answered this point above – conducting exactly the analysis requested by the reviewer. The results still hold when the maximum quantity (number or bandit 6) is excluded (Figure 2—figure supplement 1).

*Is it possible that the similarity reflected the brain being in a “magnitude evaluation” state rather than “representing magnitude 6” state between the two tasks (as one example)? If the neural encoding of specific magnitudes does not generalize across tasks, to what extent can it be said that the two tasks share a single underlying “mental number line”? Figure 3A (bottom panel), Figure 2*—figure supplement 2, *and associated analysis seem to get at this point but the link here was not entirely clear. Figure 3A (bottom panel) shows that removing the diagonal from the cross-validation across tasks did not remove the cross validation between tasks over time (that is, EEG feature vectors at one time point in the bandit task were similar to EEG feature vectors at multiple time points in the number task and vice versa). This result is interpreted to indicate that the cross-temporal similarity did not depend on shared single-quantity coding across tasks (“but by a gradual distance effect”). Could it also reflect a shared state change across tasks related to the cognitive process of quantitative comparison, rather than magnitude encoding as suggested by a shared number line? It could be useful to consider more extensively the limit to which the EEG data indicate that the neural representation of specific magnitudes generalizes across tasks.*

The reviewer makes a good point that our data could reflect in part a neural signal associated with value comparison, rather than value encoding/evaluation. However, a value comparison signal would depend on the distance between the two bandits presented on a given trial (or perhaps between successive numbers) and should mostly be observed for the second bandit, and not the first. However, we observe the magnitude signal independently and with equal strength for the first-presented and second-presented bandit (Figure 1—figure supplement 3), so we think it is unlikely to be a comparison signal. A related suggestion is that the neural signal encodes uncertainty about the choice. For example, participants may be more certain how to respond when a stimulus is 6 or 1 than when it is 3 or 4. However, we found that the strongest component of the RSA cross-validation effect placed numbers/bandits 1 and 6 at opposite ends of a neural continuum, which is at odds with this suggestion. Thus, we think the neural data we observed are more likely to be related to magnitude evaluation, rather than magnitude comparison.

Another interesting aspect of the dissimilarity data across tasks is that there seems to be greater EEG vector similarity (“confusion”) off-diagonal in the number task (Figure 1C, upper left quadrant) relative to the bandit task (lower right quadrant). Does this imply different “width” of neural tuning for magnitude in the two tasks? Although it is difficult to extrapolate from single neuron to EEG studies, single neuron recording in prefrontal and parietal cortex of monkeys performing quantity evaluation tasks such as those conducted by A Nieder, EK Miller and colleagues demonstrated that single prefrontal and parietal neurons have tuning functions for magnitude that are centered on different “preferred” quantities but are broadly tuned. This seems at least conceptually consistent with the similarity between EEG feature vectors seen off-diagonal in the upper left quadrant of Figure 1C. Conversely, similarity between EEG feature vectors seems much more concentrated along the diagonal in the bandit task (lower right quadrant, Figure 1C). This could potentially imply a neural code that is more like a look-up table in the bandit task in which a specific EEG feature vector was only seen for bandit 5 and no other bandit for example). That could suggest two magnitude representations in the two tasks, one tuned for magnitude (number task), one implementing a lookup table for magnitude (bandit task). Some comment about this potential different in magnitude representations could be useful. This would not invalidate the idea of a shared magnitude representation, but it might identify more clearly aspects in which the magnitude representation was modulated across contexts.

The reviewer is right that overall neural similarity for the numbers is greater than for the bandits. This could imply something fundamental about numerical magnitude and reward probability, but another possibility is that it has to do with differences among the way the two tasks were implemented – our numbers occurred in a rapid stream, whereas the donkeys were presented in just 2 wider-spaced intervals. This choice might seem curious but was motivated by a desire for consistency with a previous paper from our lab (Spitzer et al., 2017) where we had seen the numerical distance effect emerge strongly. Thus we are somewhat wary of trying to read too much into this difference on the basis of the current data alone. In future experiments, we plan to use more comparable tasks for numbers and bandits.

The fact that the similarity between EEG feature vectors appears much more concentrated along the diagonal in the bandit task is related to this issue. The numbers are overall more similar to each other than to the bandits, such that when a common scale is used for the 12 x 12 matrix, it appears that the EDI is stronger for the bandit task. In fact, when the data are plotted with different scales for the two quadrants, it can be seen that this is largely an artefact of the way we plotted the data. We now include a new plot that makes this clear (Figure 1C).

3) Behaviors need to be documented more carefully.The behavioral in both tasks seems very important for the research question, yet the behavior is hardly reported. The actual choices, as a function of the digits/bandits presented are not reported. In the supplementary material, Figure S3 presents some results, but these were not completely clear to me. The authors replicate the distortion in weighting numbers in the numerical task, which they have reported previously, and also provide convincing evidence that this distortion is associated with the neural signal. However, they do not show whether the behavioral distortions in the two tasks are correlated across subjects, and they do show that the neural distortions are not correlated (Figure 3B). I did not follow the logic of Figure 3C – how exactly were choice distortions in the numerical task used? Why were the distortions estimated from the bandit task regressed out? Isn't the idea that the same system is responsible for distortions in both tasks? Or do you assume separate sources for distortions, only some of which are shared across tasks?

We agree that there could have been more emphasis on the behaviour in the main text. We did in fact analyse our behaviour in some detail, but many of these analyses were relegated to the supplementary materials in order to save space (under the original “Short Report” submission). Here, in the revised paper, we have opted to return some material to the main text, and also conducted further behavioural analyses that we hope will address the reviewers’ comments. We now show the choice probabilities for both number and bandit tasks in comparable form (Figure 4A-B). The reviewer’s comments also prompted us to relate behaviour to brain activity in a new way that we find particularly revealing, by asking how differences in choice probabilities for numbers and bandits predict neural activity on each task. Interestingly, behavioural patterns for the numbers jointly explain neural distance effects in both bandit and number tasks, whereas behavioural patterns for the bandits only explain neural distances in the bandit, but not the number task (Figure 4C-D). One interpretation of this finding is that the mental number line is “primary” in that it scaffolds probabilistic reward learning, but the converse is not true. We briefly discuss why for the estimated non-linearities (Figure 4E-F), we could not find a correlation between behaviour in the numerical task and neural signals of the bandit task. It is possible our estimation method for neural RDM (‘neurometric fit’), albeit capturing certain key features, is too simple to describe the full space of the data in which there is a shared pattern.

The main thrust of the argument was that forming generalized representations of the structure of input data (such as the concept of magnitude) in one context accelerates learning in new contexts. The network model examined this question, but the behavioral data in human subjects was not interrogated to show that learning in the second task was accelerated relative to the first (unless I missed it). It is impressive that distortions in magnitude representation (biased encoding of larger magnitudes) were shared across task contexts, suggesting a shared magnitude representation. However, the behavioral or computational benefits of the underlying generalized representation in terms of accelerating learning in new contexts did not appear to be explicitly addressed. If that is the case it could be useful to consider this limitation (and potentially how it could be addressed in future studies).

Thanks for this. As discussed above, it is indeed true that our experiments do not allow us to directly assess the benefits of learning in humans. This is because we are not able to “teach” our participants the orderly/magnitude structure of symbolic numbers – they know it already. However, building on the present findings, we are conducting new experiments where we teach participants two new transitive systems, with a view to measuring exactly the transfer that is demonstrated in our simulations.

4) Some methods need to be described in more details so that the manuscript is more accessible to non-expert readers.What aspect of neural activity comprised the feature vector entered in the RSA. The Material and methods states “linear regression (GLM) was used to estimate beta coefficients for each number or (ranked) bandit for each electrode and time point.” I was puzzled what data went into the regression model and how the model was specified. Am I right that there were separate regressors for each individual number and bandit, each coded by a dummy variable, and that the regression coefficient associated with each variable (at each electrode and time point) quantified the proportion of variance in z-scored EEG signal amplitude explained by the presence (absence) of that specific quantity over stimulus repetitions (trials)?

We could have been clearer here. However, the reviewer’s description of our methods is entirely accurate. The dummy coding is a convenient way to obtain the average activation over all trials within a condition, at each electrode and time point. We have amended the Materials and methods section to make our analysis pipeline clearer.

One critical piece of information missing for me was the source of the variance the model was trying to explain (variability over stimulus presentations?) It would be important in addition to clarify whether, rather than the dummy coding above, the GLM model included numerical quantity and reward (bandit) probability as scalar independent variables (which would test a linear relation between neural activity and magnitude). Some additional explanation about the steps in the analysis, specification of the model, whether the model explained variance over stimulus repetitions (or trials) or some other source, and what the meaning of the resulting beta coefficients was in neural terms would help increase the clarity of the main finding.

Our regression-based approach tests for similarities among summary measures for each condition (e.g. bandit or number), as is standard in RSA analysis (we note in passing that this differs somewhat from classifier-based pattern similarity measures where predictions about class labels are often made at the level of individual trials). Neural summary statistics are computed condition-wise using the regression model (with dummy coding) as discussed above. It is only in the final step, when we correlate model RDMs with the neural RDMs, that we test the presence of representational patterns in the neural data. We have gone through our Materials and methods section to make sure that all of these points are clear, in particular bearing in mind that not all readers may be familiar with the standard analysis pipeline for RSA analysis.